# “ThermoTRP” Channel Expression in Cancers: Implications for Diagnosis and Prognosis (Practical Approach by a Pathologist)

**DOI:** 10.3390/ijms24109098

**Published:** 2023-05-22

**Authors:** Arpad Szallasi

**Affiliations:** Department of Pathology and Experimental Cancer Research, Semmelweis University, 1085 Budapest, Hungary; szallasi.arpad@med.semmelweis-univ.hu

**Keywords:** Transient Receptor Potential (TRP) channel, “thermoTRP”, cancer diagnosis, prognostication

## Abstract

Temperature-sensitive transient receptor potential (TRP) channels (so-called “thermoTRPs”) are multifunctional signaling molecules with important roles in cell growth and differentiation. Several “thermoTRP” channels show altered expression in cancers, though it is unclear if this is a cause or consequence of the disease. Regardless of the underlying pathology, this altered expression may potentially be used for cancer diagnosis and prognostication. “ThermoTRP” expression may distinguish between benign and malignant lesions. For example, TRPV1 is expressed in benign gastric mucosa, but is absent in gastric adenocarcinoma. TRPV1 is also expressed both in normal urothelia and non-invasive papillary urothelial carcinoma, but no TRPV1 expression has been seen in invasive urothelial carcinoma. “ThermoTRP” expression can also be used to predict clinical outcomes. For instance, in prostate cancer, TRPM8 expression predicts aggressive behavior with early metastatic disease. Furthermore, TRPV1 expression can dissect a subset of pulmonary adenocarcinoma patients with bad prognosis and resistance to a number of commonly used chemotherapeutic agents. This review will explore the current state of this rapidly evolving field with special emphasis on immunostains that can already be added to the armoire of diagnostic pathologists.

## 1. TRPs and “ThermoTRPs”: A Brief Introduction to Terminology and Biology

Capsaicin is best recognized as the active ingredient in hot chili peppers. Chili pepper is indeed “hot”, because capsaicin and noxious heat activate the very same receptor, now known as transient receptor potential, vanilloid-1, or briefly TRPV1 [1]. The molecular cloning of this receptor earned a Nobel Prize (shared with Ardem Patapoutian) for David Julius in 2021 [2].

TRPV1 is an unusual name for a receptor, and needs some explanation. Vanilloid is used because capsaicin and its ultrapotent analog, resiniferatoxin, share a vanillyl group as a motif essential for bioactivity, but differ completely in the rest of the molecule [3]. The specific binding of resiniferatoxin to a site shared by capsaicin provided the first biochemical proof of the existence of a specific “vanilloid” (capsaicin) receptor [3]. The term “transient receptor potential” (or briefly, TRP) is related to mutant fruit flies. The eye of wild-type fruit flies responds to sustained light stimuli with a lasting inward current. These mutants, however, respond to light stimuli with a transient current [4,5]. Thus, “transient receptor potential” (TRP) is really a misnomer, since the wild-type receptor, in fact, produces a lasting current. 

Based on its structural similarity to other TRP channels, the vanilloid receptor was assigned to this receptor superfamily, within which it has its own subfamilies, TRPV1 to TRPV6. The TRP superfamily has 27 mammalian members, divided into 6 subfamilies: canonical (TRPC), vanilloid (TRPV), ankyrin (TRPA), melastatin (TRPM), polycystin (TRPP), and mucolipid (TRPML) [6,7,8]. The canonical (or classical) TRP family was founded by the drosophila mutant with defective eye function. The ankyrin family (which is not really a family, because it has only one member, TRPA1) was named after the unusually long ankyrin repeat that it possesses. The first member of the melastatin family was found during a search for markers that can distinguish benign nevi from malignant melanoma. Finally, the TRPP and TRPML families were named after diseases they are associated with (polycystic kidney disease and mucolipidosis, respectively) [6,7,8]. In fact, mutant TRP channel genes are responsible for a number of diseases, so-called “TRP channelopathies” [9,10,11]. 

As TRP channel subfamilies are based on sequence homology and not function, family members often have little in common. Generally speaking, TRPs are cation channels with limited ion selectivity [7]. Some TRPs function as Ca^2+^ entry channels in the plasma membrane, whereas others regulate Ca^2+^ homeostasis in intracellular organelles, including mitochondria, Golgi network and the endoplasmic reticulum [7,11,12]. 

Cryo-electron microscopy and X-ray crystallography have provided important insights into TRP channel structure and function. In contrast to highly selective cation channels, the selectivity filter of TRPV1 is shallow and dynamic, favoring the influx of larger (e.g., Ca^2+^) or smaller (e.g., Na^+^) cations [13]. This explains the long-recognized “limited selectivity for Ca^2+^” nature of the TRPV1 channel.

At high doses, capsaicin can kill neurons by elevating intracellular Ca^2+^ levels [14]. Mitochondrial “swelling” is an early ultrastructural sign of irreversible capsaicin neurotoxicity [15]. Since TRPV1 is expressed in mitochondria [16], mitochondrial Ca^2+^ overload and resultant caspase activation may play a role in capsaicin-induced neurotoxicity. Similar, the mitochondrial TRPV1-mediated “death mechanism” may also operate in cancer cells. In keeping with this, in chronic myeloid leukemia cells, TRPV1 activation can induce apoptosis via Ca^2+^ influx, mitochondrial dysfunction, and caspase activation [17]. 

TRPV1 is not the only temperature-sensitive TRP channel, or “thermoTRP”. As of today, eleven TRP channels belonging to the TRPV, TRPM, TRPC and TRPA subfamilies have been reported to respond to thermal stimulation: in rodents, these channels cover a broad range of temperatures (Figure 1), from noxious hot (e.g., TRPV1 and TRPV2), through innocuous warm (e.g., TRPM2 and TRPV3), to cool (TRPC5) and noxious cold (TRPM8) [18]. During evolution, animals developed TRP channel orthologs with altered heat sensitivity in order to adapt to the environment in which they live. For example, camels living in desert heat express a TRPV1 protein with dramatically reduced heat sensitivity, due to a single amino acid mutation in the N-terminal ankyrin repeat [19]. 

The exact role of TRP channels in human physiological temperature sensation is still poorly understood. Historically, “thermoTRPs” have been classified as hot-, warm-, or cold-sensors. To some degree, at least under pathological conditions, this model is still applicable. For example, TRPV1-null mice display impaired noxious heat sensation in a hot plate test [20,21], and study subjects taking small molecule TRPV1 antagonists reported burn injuries as side-effects [22,23]. 

Physiological temperature sensation is a complex process. For instance, warm temperature can activate a group of sensory afferents and, at the same time, block another [24]. Furthermore, the same “thermoTRP” may respond to different temperatures, depending on the neuron in which it is expressed. There is good evidence that TRPV1 responds to noxious heat in some afferents, but detects mild temperatures in others [24]. 

It is difficult to extrapolate observations from animal experiments to humans due to marked species-related differences. A dramatic example of this phenomenon is TRPA1; this channel is a cold-sensor in mice [25,26], and a heat-sensor in frogs [27] and birds [28]. In man, the inherent thermosensitivity of TRPA1 is debated. In one study, TRPA1 responded to both heat and cold [29,30], but in another study, it lacked inherent thermal sensitivity [31]. 

During evolution, the pepper plant developed capsaicin as a chemical weapon to deter herbivores [32]. However, birds (which carry a TRPV1 ortholog that does not recognize capsaicin [33]) can eat the pepper pod and spread the pepper seed in their feces. 

Generally speaking, TRP channels are non-selective cation channels [7]. Some TRP channels such as TRPV1 display a limited selectivity for Ca^2+^, whereas others (for example, TRPM7) preferentially allow Mg^2+^ uptake [7,8,11]. In non-neuronal tissues, TRP channels have been implicated in cell growth and differentiation [34]. The participation of TRP channels in malignant transformation, cancer growth and metastasis has been reviewed elsewhere [35,36,37,38]. 

TRP channels are widely expressed in various human cancers [39,40]. Here, we review the practical use of aberrant “thermoTRP” expression, as detected by paraffin immunohistochemistry for cancer diagnosis and prognostication. 

## 2. The Expression Landscape and Function of “ThermoTRPs” in Normal Tissues

### 2.1. TRPA1

TRPA1 was originally cloned as ANKTM1, a cold-activated TRP-like channel in murine nociceptive neurons [26]. Subsequently, TRPA1 was shown to detect a broad range of irritant chemical stimuli, ranging from pleasant (such as allicin in garlic or allyl isothiocyanate in wasabi) [41] to harmful and noxious (e.g., acrolein and other electrophilic compounds in tear gas [42], cigarette smoke [43], and diesel fumes [44]). Although the thermal sensitivity of TRPA1 is markedly species-dependent (it responds to cold in mice [25,26], but it is activated by heat in birds [28]), its role as a general noxious chemical sensor is evolutionary preserved. TRPA1 is a human chemical nociception of ancient origin that first appeared in insects hundreds of millions years ago [45]. For example, the medicinal plants *Nepeta cataria* (commonly known as catnip) and *Cinnamosma fragrans* repel wild-type, but not TRPA1−/−, mosquitos [46,47], and the popular insect repellent, citronella, also acts on the TRPA1 channel both in mosquitos [48] and fruit flies [49].

The role of TRPA1 in human pain sensations is firmly established. A gain-of-function mutation in the *TRPA1* gene is responsible for familial episodic pain syndrome [50], the only known painful TRP channelopathy. This makes TRPA1 a druggable pain target. Indeed, TRPA1 is rigorously pursued by pharmacists in order to develop novel analgesic drugs [51]. 

Apart from nociceptive neurons, TRPA1 is broadly expressed in non-neuronal human tissues, ranging from the urethra [52] and bladder urothelium [53] and vascular endothelial cells [54], through keratinocytes [55], endometrial cells [56] and odontoblasts [57], to cartilage [58]. The physiological role of TRPA1 in these non-neuronal cells is largely unknown. In odontoblasts and skin keratinocytes, TRPA1 activation is believed to promote growth and differentiation. In the endothelium, TRPA1 may regulate barrier function. In the urothelium, as in nociceptive neurons, TRPA1 may respond to irritant agents in the urine [53]. 

### 2.2. TRPC5

TRPC5 was originally cloned from mouse brain [59,60]. Recombinant TRPC5 expression in HEK 293 cells potentiated ATP-induced Ca^2+^ uptake, implying its function as a store-operated cation channel [59]. In humans, TRPC5 is predominantly expressed in the brain [61], but TRPC5 expression was also demonstrated in non-neuronal tissues, including the placenta [62], gingival keratinocytes [63], odontoblasts [64], vascular endothelium and smooth muscle cells [65], and renal podocytes [66], just to cite a few examples. 

TRPC5 has been linked to nephrotic syndromes [67]. Indeed, a small molecule TRPC5 inhibitor was shown to block the progression of experimental kidney disease [68]. Furthermore, TRPC5 has been implicated in the pathomechanism of essential hypertension [69]. In the hippocampus, TRPC5 plays a central role in guiding neurite growth [70]; this implies a role for TRPC5 dysfunction in neurological disorders. In peripheral sensory neurons, TRPC5 responds to cooling [71]. Similarly, TRPC5 expressed in odontoblasts functions as a dental cold-sensor [64]. Thus, TRPC5 is an intrinsically cold-gated channel [72], with its cold-sensitivity regulated by the phosphorylation state of the channel protein.

### 2.3. TRPM2

TRPM2 was cloned from mouse brain as a cation channel activated by intracellular ADP-ribose, β-NAD^+^ or arachidonic acid [73,74,75]. Originally, this channel was called TRPC7, but later renamed TRPM2 to avoid confusion with canonical TRP channels. Recently, TRPM2 has emerged as an important cellular redox sensor [76] that regulates vulnerability to ischemic cell death during ischemic stroke [77] or cardiac ischemia–reperfusion injury [78]. 

In the central nervous system, TRPM2 has been linked to bipolar disorder and a number of neurodegenerative disorders, including Alzheimer’s and Parkinson’s disease [79]. In nociceptive neurons, TRPM2 is activated by warm temperatures [80]. Accordingly, genetic deletion of the *Trpm2* gene alters the behavioral warm sensation of animals [80,81]. Furthermore, TRPM2 is expressed in hypothalamic warm-sensitive neurons, having an important role in body temperature regulation [82].

### 2.4. TRPM3

TRPM3 was first cloned from human kidney, where it is predominantly expressed in the collecting tubules [83,84]. Expressed in HEK293 cells, TRPM3 mediated constitutive Ca^2+^ entry, which showed a profound increase when the cells were exposed to hypotonic solution [83]. These findings implied a role of TRPM3 in renal Ca^2+^ homeostasis. TRPM3 is also expressed at lesser levels in the human brain [83,85], pancreas [86], and testes [84]. 

In mice, TRPM3 is expressed in nociceptive neurons, where it is steeply activated by heat [87]. In accord, TRPM3 null animals exhibit deficits in avoiding noxious heat [87], but a complete loss of noxious heat avoidance requires a combined elimination of TRPV1, TRPA1, and TRPM3 channels [88]. Of note, gain-of-function mutations in the human *TRPM3* gene have been associated with inherited glaucoma and cataracts [89], as well as epilepsy and learning disabilities [90]. 

### 2.5. TRPM4

TRPM4 was identified by scanning the Expressed Sequence Tags database [91]. A human cDNA clone was found with significant homology to known TRPM proteins [91]. Expressed in HEK293 cells, TRPM4 functions as a Ca^2+^-activated channel [91] with marked voltage dependence [92]. In human T-cells, TRPM4 regulates Ca^2+^ oscillations [93]. In cerebral arteries, TRPM4 is expressed in smooth muscle, wherein it plays a pivotal role in maintaining myogenic tone [94]. Nitric oxide was shown to inhibit TRPM4 and thereby dilate blood vessels [95]. During autopsy, increased TRPM4 protein levels were found in vascular endothelial cells of stroke victims [96]. In the heart, TRPM4 mutations were described in conduction diseases and Brugada syndrome [97,98].

In the range of 15 to 35 C, temperature stimulates TRPM4 channel activity in inside-out membrane patches [99]. 

### 2.6. TRPM5

TRPM5 is highly expressed in the taste buds of the human tongue [100] where it plays an important role in the perception of sweet, bitter, and unami (savory) tastes [101]. Temperature has a strong influence on how we taste. In fact, cooling or heating of the tongue can be perceived as taste by many people. Thus, the finding that temperature can modify the activity of TRPM5 channels by shifting the activation curve was hardly unexpected [99]. Thermal stimuli (up to 35 C) enhance gustatory nerve responses to sweets in TRPM5 wild-type, but not in TRPM5-null, mice [99] 

In humans, TRPM5 expression has also been demonstrated in hair follicles [102], the lacrimal sac epithelium [103], pancreatic β-cells [104], and sinonasal mucosa [105], as well as in the gastro-intestinal tract (stomach, small and large intestine) [106,107]. The glucose intolerant phenotype of the TRPM5 (−/−) mouse implies a role for aberrant TRPM5 activity in the pathomechanism of type-2 diabetes [108]. 

### 2.7. TRPM8

In 2002, TRPM8 was cloned independently in the laboratory of the two recipients of the 2022 Nobel Prize in Physiology and Medicine, David Julius [109] and Ardem Patapoutian [110], as a channel that senses cold stimuli and menthol. 

In humans, TRPM8 mRNA was first detected in the male genito-urinary tract (the prostate, testicle, seminiferous tubules, scrotal skin, and urinary bladder) [111]. Subsequently, functional TRPM8 was found in lung epithelial cells [112], odontoblasts [113], adipocytes [114], and corneal endothelial cells [115]. TRPM8 is a well-established, druggable target for dry eye disease [116] and cold-hyperalgesia [117]. 

Clinical trials with TRPM8 agonists for itch (Cryosim-3 gel, Phoenix Pharma, Burlingame, CA, USA) [118] and cough relief (AX-8, Axalbion, Manchester, UK) [119] are ongoing. The TRPM8 agonist, D3263 (Dendreon, Seal Beach, CA, USA), was also trialed in a limited number of patients with solid tumors, including advanced prostate cancer [120]. 

### 2.8. TRPV1

As discussed above, TRPV1 was originally cloned as the capsaicin receptor [1]. The three cardinal activation modes of TRPV1 are capsaicin (and other vanilloids), protons, and heat [121,122]. In mammals, TRPV1 also functions as a shared receptor for painful venoms and toxins, such as those present in spiders [123] and jellyfish [124]. As expected, TRPV1 is highly expressed in primary sensory neurons (in fact, TRPV1 was cloned from a sensory neuron cDNA library [1]). Unexpectedly, TRPV1 is also expressed, albeit at much lower levels, in brain nuclei [125], as well as in various non-neuronal tissues, ranging from keratinocytes [126] and immune cells [127] to vascular smooth muscle [128]. In human skin, TRPV1 is the predominant “thermoTRP” (Figure 2). In addition to its pivotal role in nociception, TRPV1 has been implicated in thermoregulation [129], diabetes [130], appetite control [131], and blood pressure regulation [132]. 

There is a large body of literature on the oncogenic role of TRPV1, with conflicting results. There is, however, good evidence that TRPV1 is expressed in both sensory afferents and immune cells in the tumor microenvironment, as well as in the tumor cells themselves [134]. This expression pattern may create an intricate, and as yet poorly understood, interaction between cancer cells, nerves, and immune cells. In fact, chemical ablation of TRPV1-positive afferents by capsaicin [135] or resiniferatoxin [136] has a profound effect on tumor growth and metastasis; for example, it caused early metastatic spread in a murine model of triple negative breast cancer [135]. By contrast, it prolonged (tripled) the survival of mice inoculated with B16F10 melanoma cells [136]. 

### 2.9. TRPV2

Cloned as a capsaicin receptor homologue (vanilloid receptor-like protein-1, VRL-1) with a high threshold for noxious heat [137], TRPV2 is predominantly expressed in the brain and dorsal root ganglia. TRPV2 is also expressed in aortic myocytes where, at least in mice, it responds to osmotic changes [138]. In human lens epithelial cells, TRPV2 is activated by a high-glucose environment, and the resultant Ca^2+^ accumulation leads to cell death (presumably a mechanism of diabetic cataract) [139]. In addition, TRPV2 expression was reported in endometrial stromal cells [140], endothelial cells and cardiomyocytes [141], just to cite a few examples. Indeed, TRPV2 is a remarkably conserved protein, expressed in almost all human tissues studied [142].

### 2.10. TRPV3

TRPV3 was identified independently by three groups as a temperature-sensitive (22 to 40 °C) cation channel, predominantly expressed in the skin, brain, spinal cord, and sensory ganglia [143,144,145]. In the skin, TRPV3 has been implicated in various functions. Gain-of-function point mutations in the *TRPV3* gene are responsible for a debilitating skin condition known as Olmsted syndrome [146]. Furthermore, abnormal TRPV3 activity may cause atopic dermatitis [147] or hair loss [148]. Recently, TRPV3 was found in both the small and large intestine, with reduced expression in inflammatory bowel disease [149]. TRPV3 as a therapeutic target has been reviewed elsewhere [150].

### 2.11. TRPV4

TRPV4 (formerly, OTRPC4) was originally identified in kidney, liver and heart as non-selective cation channel with remarkable sensitivity to changes in volume [151] and extracellular osmolarity [152]. Indeed, TRPV4 null mice showed impaired osmotic sensation [153]. Disrupting the *Trpv4* gene in mice also reduced the pressure-sensitivity, leaving heat sensation intact [154]. Based on these observations, an essential role of TRPV4 in normal osmotic sensation and pressure detection was postulated. 

In mouse keratinocytes, TRPV4 responds to modest increases in ambient temperature [155]. It was speculated that thermal activation of TRPV4 in human keratinocytes evokes the itchy feeling in rosacea [156]. TRPV4 is also expressed in human sperm, in which it guides migration towards the warm womb [157]. 

Functional TRPV4 expression was reported in human airway smooth muscle cells, macrophages, oral and vaginal keratinocytes, urothelial cells, cardiac myocytes, etc [158]. Point mutations in the *hTRPV4* gene (so-called “TRPV4 channelopathies”) have been linked to severe skeletal dysplasias and neuromuscular disorders, including brachyolmia and Charcot–Marie–Tooth disease [159,160]. 

## 3. Aberrant “ThermoTRP” Expression in Cancers: Implications for Diagnosis and Prognostication

Pathologists rely on immunostains performed on paraffin-embedded tissues to determine the lineage of the tumor. For example, carcinomas are positive for cytokeratins, hematopoietic malignancies express CD45, and melanoma shows Melan-A and/or HMB45-like immunoreactivity. Unfortunately, these stains do not distinguish between benign and malignant lesions. 

Pathologists also use immunostains to provide information on the prognosis (for example, “triple-negative” breast cancer usually follows an aggressive course, whereas ALK-positivity portends a favorable prognosis in anaplastic large cell lymphoma [161]), or guide clinical treatment decisions. For example, a number of selective BRAF inhibitors (e.g., vemurafenib and dabrafenib) are available for patients with metastatic melanoma [162]. Her2/neu-positive breast cancers react to targeted therapy with trastuzumab [163], whereas CD20- or CD30-positive lymphomas can be treated with the humanized monoclonal antibodies rituximab [164] and brentuximab [165], respectively. BRAF, Her2, CD20 and CD30 can be easily detected by paraffin immunostains. 

The antibodies that we use to detect these proteins in everyday practice are well characterized, and their staining methods standardized. Unfortunately, many broadly used anti-“thermoTRP” protein antibodies lack specificity. For example, of the five tested commercially available anti-TRPA1 antibodies, only two proved selective for TRPA1 [166]. Therefore, the previously published data regarding human TRPA1 expression in normal and cancerous tissues should be revisited. Similar concerns have been raised about the specificity of anti-TRPV1 antibodies [167].

*a*.
*Squamous cell carcinoma of the skin and the head-and-neck*


TRPV1 is highly expressed in human skin keratinocytes (Figure 2), predominantly in membranous staining patterns (Figure 3). TRPV1-like immunoreactivity is increased in human oropharyngeal squamous cell carcinoma (SQCC) and skin SQCC cases (Figure 4), compared to control tissue [168,169,170]. In normal human oral mucosa, TRPV1-like immunoreactivity is restricted to the stratum basale, whereas in cancer, it is present throughout the whole epithelium [168]. Interestingly, in patients with a long history of smoking and/or alcohol abuse, TRPV1 staining (similar to cancerous tissue [168]) can also be seen in keratinocytes above the basal layer [171]. The prognostic value of TRPV1 immunostaining in oral SQCC is unknown. Parenthetically, one study described similar TRPV1-like immunostaining between healthy controls and human skin SQCC samples [172]. The cause of this discrepancy is unknown. In part, it may be related to the antibody used in the study. As mentioned above, the specificity of some anti-TRPV1 antibodies is questionable [167]). 

In addition to SQCC, TRPV1 is highly expressed in basal cell carcinoma (BCC; Figure 5) [169]. TRPV4-like immunoreactivity was also demonstrated in invasive skin SQCC, along with two acid-sensitive ion channels, ASIC1 and ASIC2 (Figure 4) [169]. 

Functional TRPA1 expression was demonstrated in nasopharyngeal SQCC, using a combination of immunohistochemistry, functional (TRPA1 agonist-induced Ca^2+^-uptake), in situ hybridization (RNAScope), and molecular studies (qPCR), at levels much higher than in healthy oral mucosa samples [170]. Of all the TRP channels, the *TRPA1* gene shows the highest expression in head-and-neck SQCC [40]. In fact, the *TRPA1* gene is part of the 12-gene methylation signature panel (REASON score) that predicts adverse clinical outcome in patients with an early stage of oral SQCC [173]. 

Functional TRPM8 expression was reported in two human SQCC cell lines derived from tongue cancer, HSC3 and HSC4 [174]. TRPM8 activation was augmented by menthol, whereas the small molecule TRPM8 antagonist RQ-00203078 blocked the migration of cancer cells in gelatin [174]. 

TRPM2-like immunoreactivity was also described in human tongue SQCC [175]. In this study, TRPM2 staining was virtually absent in the controls. If this finding is verified by other investigators, TRPM2 may be a novel immunohistochemical marker to distinguish between reactive and malignant oral mucosa lesions. 

The *TRPC4* gene is highly expressed in head-and-neck SQCC [40]. Strong TRPC4-like immunoreactivity was demonstrated in skin SQCC, but not in BCC [176]. Therefore, the BerEP4 (positive in BCC and negative in SQCC)–TRPC4 (negative in BCC and positive in SQCC) combination may be useful in the differential diagnosis of BCC and SQCC cases. 

*b*.
*Pulmonary small cell carcinoma and adenocarcinoma*


TRPV1 expression in pulmonary adenocarcinoma portends adverse prognosis [177,178,179] and predicts resistance to certain chemotherapeutic agents such as cisplatin or 5-fluorouracil [180]. The prognostic significance is based on the measurement of TRPV1 mRNA in tumor and control lung tissues, and is yet to be validated by immunostaining. 

TRPV3 immunostaining may also identify a subset of patients with bad prognosis [181]. High TRPV3 levels were detected in 68% of the cancer cases tested (65 out of 96 patients) [181]. Importantly, TRPV3 expression inversely correlated with cancer differentiation [181]. 

By contrast, in a cohort of 95 patients with lung adenocarcinoma, TRPC3 expression as determined by mRNA levels (real-time RT-PCR) identified a group with good prognosis [182]. In this study, immunostains were performed with an anti-TRPC3 antibody (cat: 54616, AnaSpec, San Jose, CA, USA); the tumor cells showed strong cytoplasmic staining, whereas control pneumocytes were either negative or weakly positive. This study implies both the diagnostic and prognostic value of TRPC3 immunohistochemistry in the work-up of suspected lung adenocarcinoma. 

In four human small cell carcinoma cell lines (H69, H146, H187 and H510), high TRPA1 mRNA expression was demonstrated using RT-PCR [183]. In these cells, TRPA1 activation promoted tumor cell survival and growth. This is interesting because small cell carcinoma is a disease of smokers, and TRPA1 is a well-established target for irritant compounds in cigarette smoke [43]. Furthermore, pulmonary small cell carcinoma has a dismal prognosis and few effective therapeutic options. Therefore, it may be worth exploring if TRPA1 can be targeted by antagonists to halt (or at least slow) the progression of small cell carcinoma. 

*c*.
*Prostate cancer*


In prostate carcinoma, TRPV2 expression has been associated with the aggressive, castration-resistant phenotype [184]. Accordingly, no TRPV2 expression was observed in pT2 tumors; TRPV2 occurred only in advanced tumors with metastatic disease [184]. 

TRPM8 is the predominant “thermoTRP” in normal prostate (Figure 6). In prostate cancer, TRPM8 expression shows a strong correlation with grade (ISUP grade 4 or higher) and perineural invasion (Figure 7) [185]. Of note, circulating TRPM8 mRNA is a molecular signature of high-risk disease [186]. Taken together, these studies suggest that TRPV2 and TRPM8 (along with NKX3.1) may constitute a valuable immunohistochemical panel to diagnose prostate cancer, and may identify patients at risk of aggressive disease who need early therapeutic intervention (as opposed to watchful waiting). 

In principle, TRPM8-positive cancer cells can be found and visualized in the body by radiohalogen ligands [187]. This finding may open a new window of opportunity for detecting metastatic prostate cancer. TRPM8 is a promising therapeutic target in advanced prostate cancer [188,189]. In fact, clinical trials with D3263 have already been completed in a small number of patients [120], the outcome of which is yet to be disclosed. 

Using in situ hybridization, TRPV6 could not be detected in benign prostatic tissue (including benign prostatic hyperplasia), prostatic intraepithelial neoplasia (high-grade PIN), or small, incidental adenocarcinoma [190]. In a study of 96 prostatectomy specimens, TRPV6 mRNA transcript levels were positively correlated with Gleason/ISUP score, extraprostatic extension, and lymph node metastasis [190]. If these observations can be validated by immunohistochemistry, TRPV6 may be another useful surrogate marker of aggressive disease. 

*d*.
*Bladder cancer*


TRPV1 protein is easily detectable in the normal urothelium (Figure 8a). In non-invasive papillary urothelial carcinoma, TRPV1 expression is reduced (but still detectable) compared to normal urothelium [191], whereas in invasive urothelial carcinoma, TRPV1 staining is virtually absent (Figure 8b) [192,193]. According to these observations, TRPV1 immunostaining may help distinguish between non-invasive and invasive urothelial carcinoma. Moreover, Kaplan–Meier curves demonstrated a significantly shorter survival for patients with TRPV1 mRNA downregulation [193]. Thus, the absence of TRPV1-like immunoreactivity may have an independent negative prognostic significance in patients with bladder cancer. 

The normal human urothelium expresses TRPV2 protein (detected by the goat anti-human TRPV2 polyclonal antibody, Santa Cruz Biotechnology) in the superficial layer, mostly in umbrella cells [194]. By contrast, in urothelial carcinoma, strong and uniform nuclear and cytoplasmic staining was seen throughout the tumor [194]. The TRPV2 immunoreactivity score correlated with the stage of the cancer.

In a retrospective study of 156 archived paraffin-embedded urothelial carcinoma cases, strong and uniform TRPM8-like immunoreactivity was detected in 54% of the cancers; the matched non-cancerous tissue samples showed lower intensity staining in scattered cells [195]. A Kaplan–Meier curve analysis indicated a shorter overall survival time for patients with strong TRPM8 staining [195].

*e*.
*Breast cancer*


TRPV1 is expressed in normal breast tissue (Figure 9). In invasive ductal carcinoma, three distinct TRPV1 staining patterns have been described using the Abcam (Cambridge, MA, USA) anti-TRPV1 antibody: “classical” (diffuse staining in membrane and cytology), “non-classical” (endoplasmic reticulum/Golgi pattern), and “mixed” (Figure 10) [196]. The classical pattern was predominantly seen in Luminal A and B cancers, whereas the non-classical pattern has been associated with Her2-positive and triple-negative (BCL-like) breast cancer and adverse clinical outcome [196]. The inter-observer variability of this staining pattern recognition is yet to be determined. Of note, TRPV1 expression in breast cancer cell lines and animal models has a large body of literature which is beyond the scope of this review. 

In triple-negative breast cancer, TRPV2 expression seems to identify a group of patients with a favorable prognosis [197]. 

In a study of 59 women with invasive ductal carcinoma, TRPV6 expression was found to be elevated compared to both adjacent non-cancerous tissue and ductal carcinoma in situ (DCIS) [198]. TRPV6 expression was also associated with metastatic disease [199]. 

In breast cancer biopsy tissues, TRPC3 and TRPC6 proteins (determined by Western blotting) were upregulated compared to normal breast tissue [200]. This study is yet to be validated with paraffin immunohistochemistry. 

TRPM8 was also reported to be overexpressed in breast carcinoma, having a positive correlation with the mitotic (Ki67) index and the Scarff–Bloom–Richardson grade [198].

*f*.
*Gastric adenocarcinoma*


Using the Abcam anti-TRPV1 antibody (cat: ab3487), markedly reduced (or virtually absent) TRPV1 protein expression was found in gastric adenocarcinoma [201]. Strong TRPV2 [202] and/or TRPM8 [203] protein expression was found in a subset of patients with adverse outcomes (shorter overall survival). The TRPV2 study involved a total of 1524 cancer samples both at the mRNA and protein level. Furthermore, TRPV2 expression is predictive of resistance to cisplatin therapy [204], which may provide a partial explanation for the negative predictive value of the TRPV2 expression. TRPV4 expression has been associated with early lymph node metastasis and poor overall survival [205]. 

*g*.
*Colorectal adenocarcinoma*


With immunohistochemistry using an anti-TRPV1 antibody (Cell Signaling Technologies, Boston, MA), decreased TRPV1 protein expression was found in cancer biopsies compared to adjacent normal tissue [206].

In a study of 93 patients, decreased TRPV3 and TRPV4 mRNA was found in colonic adenocarcinoma compared to normal tissue [207]. Using RT-PCR, increased TRPM8 mRNA expression was found to render negative prognostic value [208]. Unfortunately, these studies did not include paraffin immunostains; therefore, they cannot be applied to routine pathology. 

*h*.
*Pancreatic ductal adenocarcinoma*


Both TRPM7 and TRPM8 proteins are absent in normal pancreatic ducts [209], but are present in a subset of pancreatic ductal adenocarcinoma patients [209,210,211,212], where TRPM8 expression heralds adverse outcomes [213]. Since TRPM8 is also expressed in a broad range of adenocarcinomas (such as breast [198] and stomach [203]), TRPM8-positivity lacks specificity for determining the primary site of the cancer. 

In human pancreatic adenocarcinoma cell lines, robust functional TRPA1 expression was demonstrated [214]. This is yet to be verified in actual human tumor samples. 

*i*.
*Endometrial and ovarian carcinoma*


Of all the 27 human TRP channel genes examined, TRPV2 shows the highest expression in endometrial carcinoma [40], where elevated TRPV2 mRNA expression heralds an adverse outcome [215,216]. These mRNA studies need to be correlated with TRPV2 immunostaining. 

In ovarian carcinoma, TRPV1 expression is increased compared to normal control or borderline lesions [217]. Patients with high TRPV1 protein and low pTEN expression seem to have especially bad prognosis [217]. This extensive study involved 217 carcinoma patients and 157 benign ovarian tumors. TRPV1-immunoreactivity was determined by the anti-TRPV1 polyclonal antibody raised in rabbits (Alomone, Jerusalem, Israel; cat ACC-030).

According to an analysis of The Cancer Genome Atlas and Genotype-Tissue Expression databases, strong TRPV4 expression predicts multidrug resistance and resultant adverse outcomes in ovarian carcinoma [218]. Again, this study needs to be confirmed with paraffin immunohistochemistry. 

*j*.
*Renal cell carcinoma*


Of human TRP genes, TRPM2 shows the highest expression in conventional (clear cell) renal cell carcinoma (ccRCC) [40]. A second study analyzed TRPM2 expression in the Tumor Immune Estimation Resource (TIMER) and Gene Expression Profiling and Interactive Analysis (GEPIA) databases: TRPM2 mRNA was elevated in ccRCC compared to non-neoplastic kidney, and TRPM2 upregulation predicted poor survival [219].

TRPV1 protein expression is strong in normal renal tubules (goat polyclonal anti-TRPV1 antibody, Santa Cruz, CA, USA), whereas it is diminished or lost in ccRCC [220]. The loss of TRPV1 staining correlated with the Fuhrman grade of the tumor and predicted poor survival [220]. 

*k*.
*Hepatocellular carcinoma*


Normal liver is devoid of TRPV1 staining. By contrast, TRPV1-like immunoreactivity was detected in 81% of the cirrhosis, and 48% of the hepatocellular carcinoma cases [221]. Patients with TRPV1-positive carcinoma had a better prognosis [221]. 

Increased TRPV2 protein expression was found in 84% of cirrhosis cases compared to normal liver [222]. In this study, 29% of hepatocellular carcinoma cases showed high TRPV2 protein expression; these were predominantly poorly differentiated cancers with evidence of portal vein invasion [222]. Taken together, these findings imply that a combination of TRPV1 and TRPV2 immunostains may separate low risk (TRPV1+/TRPV2−) and high-risk (TRPV1−/TRPV2+) hepatocellular carcinoma patients. 

In hepatocellular carcinoma, increased TRPV4 protein and mRNA levels were found compared to paired non-tumoral liver tissue [223]. The prognostic significance of TRPV4 expression in liver cancer is, however, unclear.

*l*.
*Melanoma*


TRPM1 (also known as melastatin) plays a pivotal role in normal melanocyte pigmentation, and its expression positively correlates with melanin content [224]. Although TRPM1 expression does not reliably distinguish between benign nevi and malignant melanoma, the loss of TRPM1 mRNA in melanoma predicts metastatic disease and poor survival [225]. In fact, quantification of TRPM1 mRNA by chromogenic in situ hydridization (CISH) revealed a steep TRPM1 loss at the transition of the melanoma from the radial growth phase into the vertical growth phase, with adverse prognostic significance [226,227]. This observation implies that TRPM1 CISH may help differentiate between in situ and tumorigenic melanoma cases. TRPM1 CISH may also help distinguish Spitz nevi from melanoma; complete absence of TRPM1 mRNA was observed in 27 out of 33 (82%) of melanomas, but only 1% (1 in 95) of Spitz nevi [228]. 

Though beyond the scope of this review, it is worth mentioning that autoantibodies are responsible for the melanoma-associated retinopathy target TRPM1 cation channel of retinal ON bipolar cells [229]. TRPM1 was also identified as a potential risk gene (along with 35 other genes) in familial melanoma [230].

TRPM8 is expressed in the human melanoma cell line, G361 [231]. This is of interest because TRPM8 is an established and already clinically pursued oncotarget. TRPV1 protein expression was demonstrated in melanoma (Figure 11), but it was also detected in benign nevi (Figure 12); therefore, it cannot differentiate between benign and malignant melanocytic proliferations [169].

In the Cancer Genome Atlas database, TRPM4 and TRPV2 were identified as negative prognostic markers in uveal melanoma. TRPV2-positive cases had particularly dismal prognoses; over half of the patients died within one year of the diagnosis with metastatic disease [232]. 

*m*.
*Gliomas, including glioblastoma multiforme (GBM)*


In a study of 33 patients with GBM, gene expression profiling identified significant increases in the expression level of several “thermoTRP” genes, including TRPM2, TRPM3, TRPM8, TRPV1 and TRPV2 [233]. 

Using paraffin immunohistochemistry, TRPV1 and TRPA1 protein expression was found in 62% of the WHO grade II astrocytomas, 37.5% of the anaplastic astrocytomas (WHO grade III), and 16.3% of GBM cases [234]. Another study using immunofluorescence also demonstrated TRPV1-like immuoreactivity in high-grade gliomas [235].

*n*.
*Hematolymphoid malignancies*


The expression pattern of “thermoTRP” genes in hematological malignancies, including leukemias, lymphomas, and plasmacell neoplasms, has been studied extensively using molecular approaches [236,237,238]. For example, there is emerging evidence that TRPV2 expression may be an independent negative prognostic marker in plasma cell myeloma [238]. However, these observations have to be verified with paraffin immunohistochemistry before they can be introduced into the practice of diagnostic hematopathology. 

## 4. Conclusions and Future Research Directions

With over a thousand research papers and 22 reviews, the literature on TRP channels and cancer is vast. The complete literature is probably even larger, since many relevant studies had been published before the term TRP channel was introduced. For example, the archetypal “thermoTRP” channel is the capsaicin receptor, TRPV1 [1]. Using the keywords capsaicin and cancer, PubMed lists close to a thousand papers published since 1978. 

The role of TRP channels, including temperature sensitive “thermoTRP” proteins, in malignant transformation, tumor growth and metastasis has been the subject of excellent reviews [35,36,37,38,38,239,240,241,242,243,244,245]. Briefly, TRP channels are expressed both in cancer cells and in the tumor microenvironment, including nerves, blood vessels and immune cells [134]. Many TRPs function as Ca^2+^ channels, and dysregulated intracellular Ca^2+^ has been implicated in carcinogenesis [246]. Furthermore, TRP channel expression in sensory afferents and immune cells is thought to create an intricate, and as yet poorly understood, neuro-immune network that can impact the survival, proliferation, and metastatic spread of cancer. In keeping with this concept, chemical ablation by resiniferatoxin of sensory afferents has been shown to accelerate the growth of experimental breast carcinoma [135], and, conversely, inhibit the progression of melanoma [136]. 

TRP channels, as promising oncotargets, have also been reviewed elsewhere [247,248]. For example, human nasopharyngeal squamous cell carcinoma cells express functional TRPA1 and TRPV1 channels [170]. In vitro, TRPA1 and/or TRPV1 activation has been shown to kill squamous cell carcinoma cells [170]. These observations imply a therapeutic value of TRPA1 and/or TRPV1 agonists in the management of inoperable nasopharyngeal squamous cell carcinoma. TRPM8 is an established and already clinically pursued target in cancer therapy. The TRPM8 agonist, D3263 (Dendreon, Seal Beach, CA, USA), has already been trialed in a limited number of patients with solid tumors, including advanced prostate cancer [120]. 

The wide distribution of “thermoTRPs” [133] suggests a diverse function beyond heat sensation. For example, the mechanosensitive TRPA1 [249] and TRPV4 [250] channels are expressed in the gastrointestinal tract, implicating them in motility disorders [251]. In fact, TRPA1 activation can stimulate bowel motility [252]. Therefore, TRPA1 agonists may be clinically useful in postoperative ileus [252] and atonic colon/chronic constipation, also known as “lazy bowel syndrome”. By contrast, TRPA1 antagonism may relieve colic pain with added antidiarrheal activity [251]. Even TRPV1, long considered to be a marker of nociceptive neurons [253], is detectable in a wide range of tissues, ranging from keratinocytes [126] and melanocytes [169] to glia [235] and lymphocytes [127]. As predicted by this tissue distribution, TRPV1 expression has been described in carcinomas [168,169,170,171,177,178,179], melanomas [169], gliomas [235], and hematological malignancies [236,237,238]. No “thermoTRP” examined so far has had acceptable specificity in determining the lineage of a tumor.

A number of “thermoTRPs” may aid pathologists in distinguishing between benign and malignant lesions (Table 1). For example, TRPV1 is expressed in normal or inflamed gastric mucosa, but is absent in gastric adenocarcinoma [201]. Furthermore, TRPV1 is expressed both in the normal urothelium [254] and non-invasive papillary urothelial carcinoma [191]. By contrast, no TRPV1 expression was seen in invasive urothelial carcinoma [192,193]. Although not temperature-sensitive, TRPM1 is worth mentioning, since TRPM1 mRNA CISH may distinguish between melanoma and Spitz nevi [228]. Melanomas with strong and uniform TRPM1 mRNA CISH positivity are aggressive, commonly with early metastasis and death [225]. 

The use of “thermoTRP” immunohistochemistry to predict good or dismal clinical outcomes looks promising (Table 2). A good example is TRPM8 expression in prostate cancer. TRPM8 expression predicts aggressive behavior with early metastatic disease and adverse prognoses [185]. TRPM8 expression can also identify patients who may be potential candidates for future clinical trials with TRPM8 antagonists [120,188,189]. Another example is TRPV1. TRPV1 expression can dissect a subset of pulmonary adenocarcinoma patients with bad prognosis [177,178,179] and resistance to a number of commonly used chemotherapeutic agents such as cisplatin and 5-fluorouracil [180]. 

Despite the extensive literature on “thermoTRP” expression and cancer, very few comparative studies are available. For example, we know that pulmonary adenocarcinoma is strongly positive for TRPC3 [182], but we do not know whether or not other adenocarcinomas are also TRPC3-positive. Another big problem is the questionable specificity of some commonly used anti-TRP antibodies, as exemplified by TRPA1 [166], TRPV1 [167], and TRPM8 [255]. It is possible that the literature is littered with reports of non-specific immunostaining. A growing number of papers are analyzing public cancer genome databases to find TRP channels with prognostic potential [39,40,218,219,232]. These findings have to be correlated with paraffin immunostains.

## Figures and Tables

**Figure 1 ijms-24-09098-f001:**
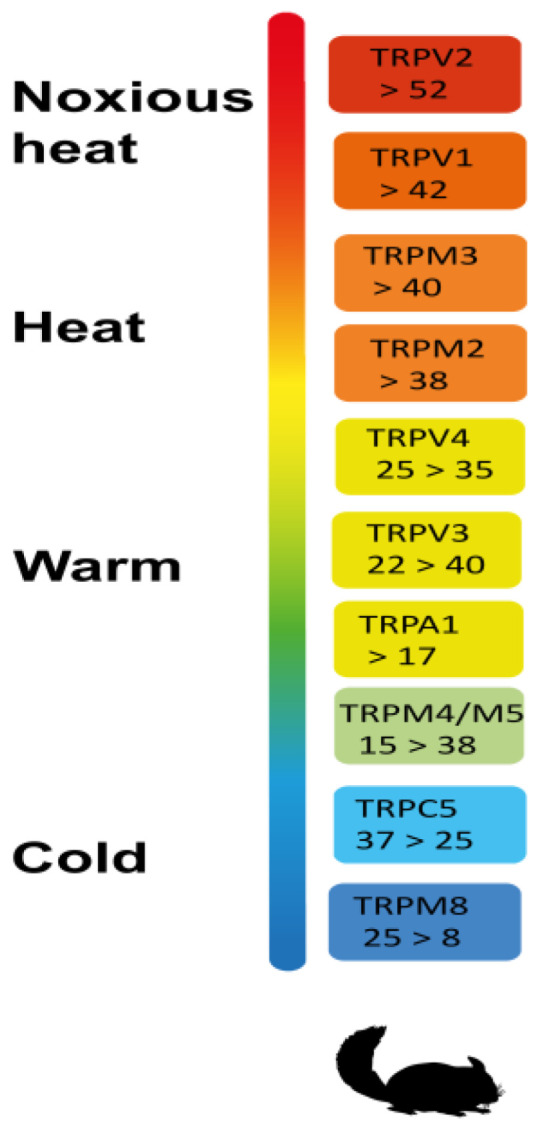
The temperature-sensitive TRP channels, so-called “thermoTRPs”.

**Figure 2 ijms-24-09098-f002:**
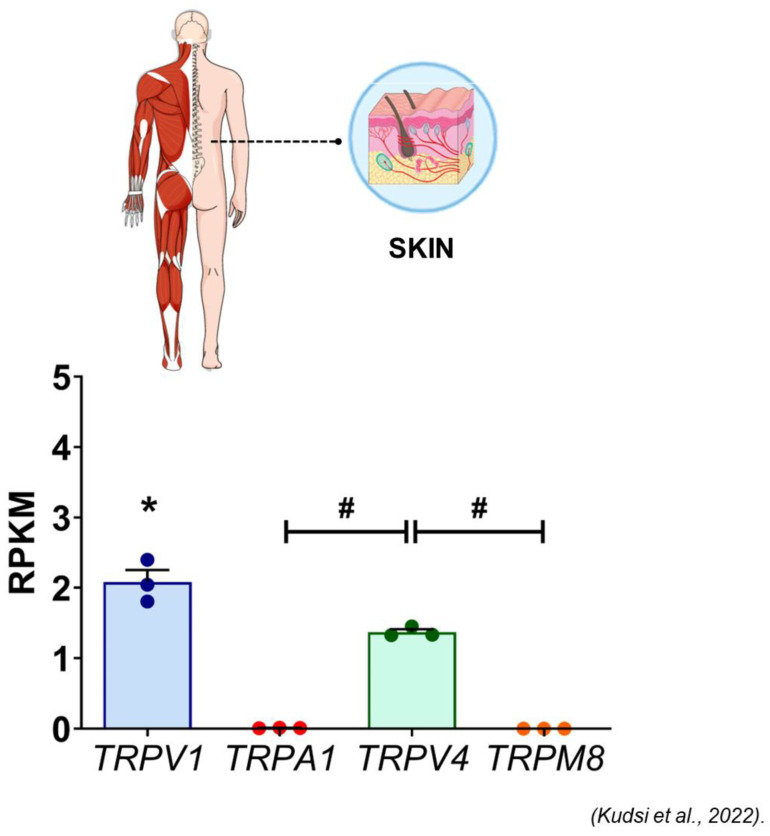
The relative transcription of genes encoding TRPV1, TRPA1, TRPV4 and TRPM8 in human skin (reproduced with permission from [133]). The results are expressed as mean + S.E.M. and analyzed by one-way ANOVA followed by the Tukey post hoc test. * indicates *p* < 0.05, significant for 3 genes; # indicates *p* < 0.05, significant or 1 or 2 genes.

**Figure 3 ijms-24-09098-f003:**
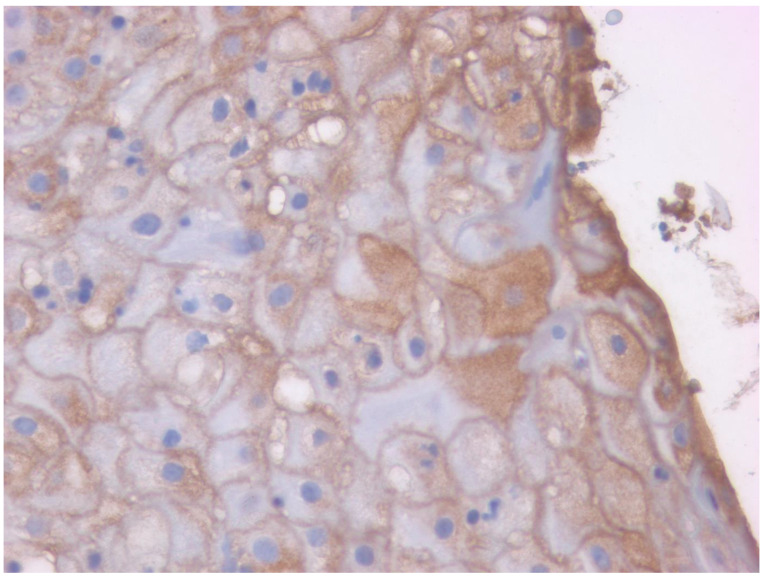
TRPV1-like immunoreactivity in human skin. Image captured at 40× magnification.

**Figure 4 ijms-24-09098-f004:**
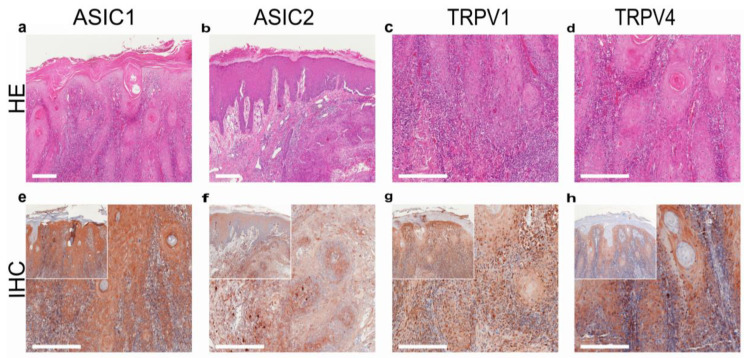
Squamous cell carcinoma of the human skin: TRPV1- and TRPV4-like immunoreactivity, along with two major acid-sensitive channels, ASIC1 and ASIC2 (reproduced from [164]). (**a**–**d**) H&E staining of tumor samples; (**e**–**h**) immunohistochemical staining (inserted smaller pictures a 2× larger perspective). Scale bars represent 200 µm.

**Figure 5 ijms-24-09098-f005:**
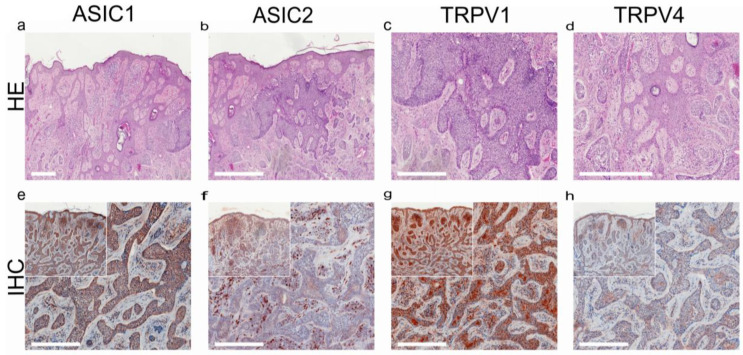
Basal cell carcinoma of the human skin: TRPV1- and TRPV4-like immunoreactivity, along with two major acid-sensitive channels, ASIC1 and ASIC2 (reproduced from [164]). (**a**–**d**) H&E staining of tumor samples; (**e**–**h**) immunohistochemical staining (inserted smaller pictures a 2× larger perspective). Scale bars represent 200 µm.

**Figure 6 ijms-24-09098-f006:**
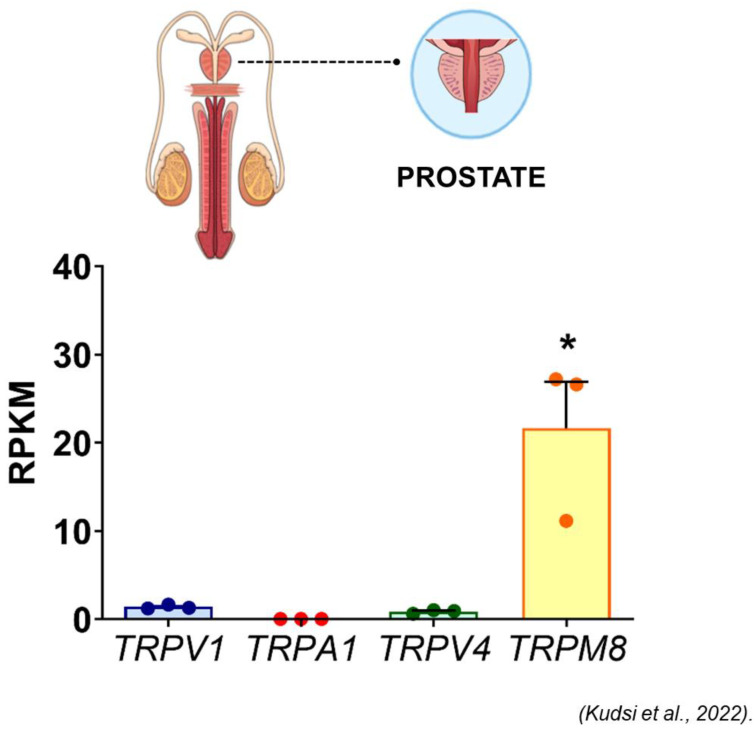
The relative transcription of genes encoding TRPV1, TRPA1, TRPV4 and TRPM8 in the human prostate (reproduced with permission from [133]). The results are expressed as mean + S.E.M. and analyzed by one-way ANOVA followed by the Tukey post hoc test. * indicates *p* < 0.05, significant for 3 genes.

**Figure 7 ijms-24-09098-f007:**
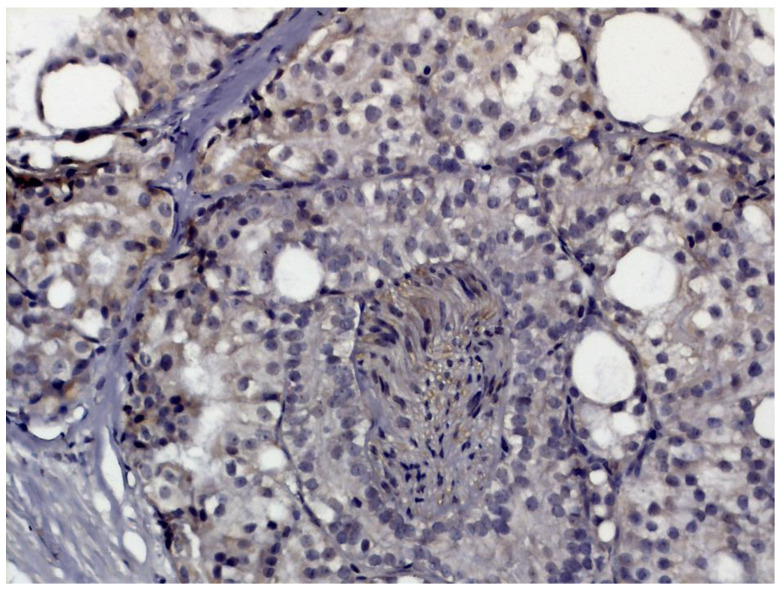
TRPM8-like immunoreactivity in prostatic adenocarcinoma with perineural invasion (reproduced with permission from [180]). Image captured at 40× magnification.

**Figure 8 ijms-24-09098-f008:**
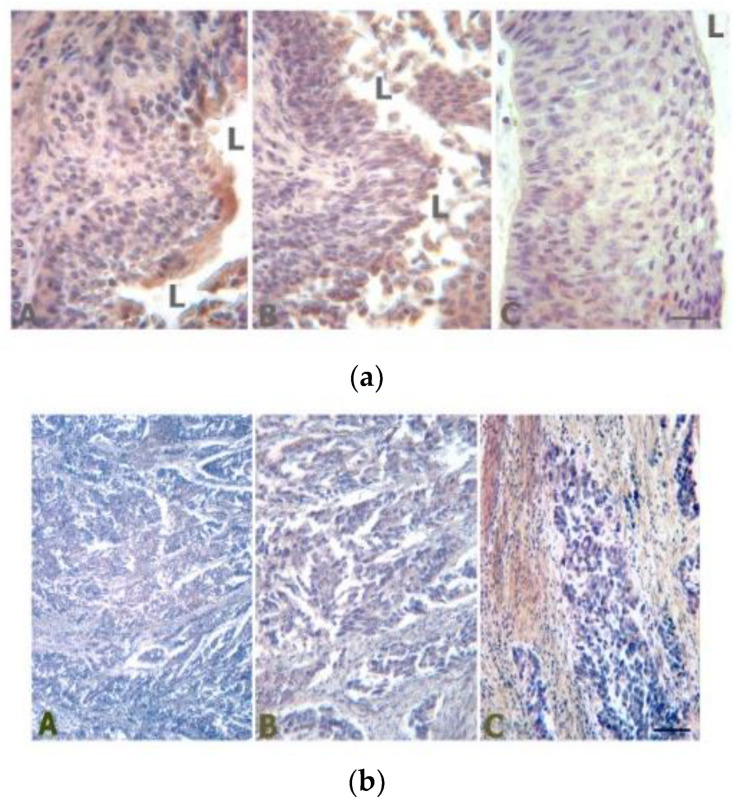
(**a**) TRPV1-like immunoreactivity in a normal human urothelium and non-invasive urothelial carcinoma (reproduced from [188]). L-lumen of the bladder; A-normal urothelium; B-papillary urothelial carcinoma; C-in-situ urothelial carcinoma; (**b**) TRPV1-like immunoreactivity is absent in invasive urothelial carcinoma. Image captured at 10× magnification.

**Figure 9 ijms-24-09098-f009:**
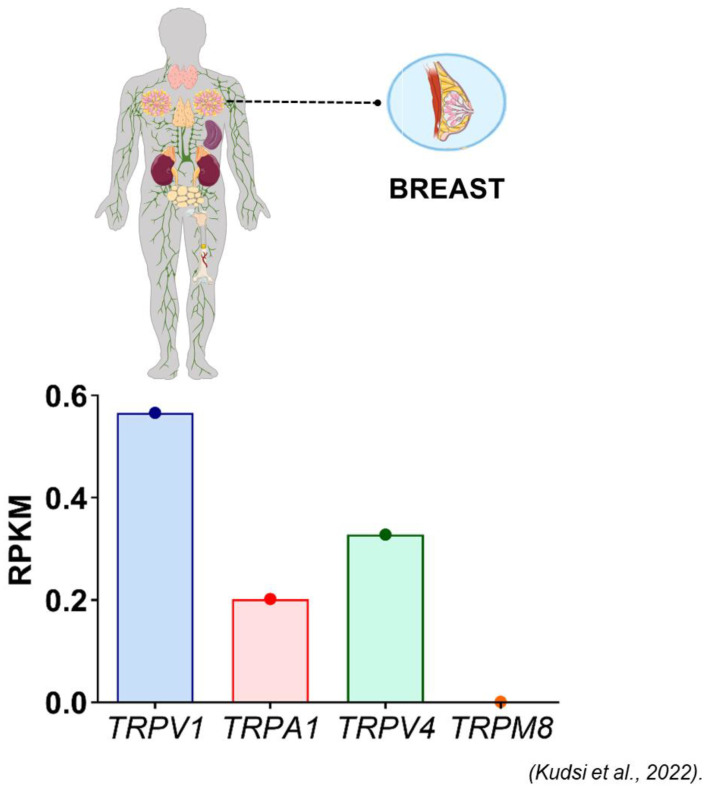
The relative transcription of genes encoding TRPV1, TRPA1, TRPV4 and TRPM8 in the human female breast (reproduced with permission from [133]). The results are expressed as mean + S.E.M. and analyzed by one-way ANOVA followed by the Tukey post hoc test.

**Figure 10 ijms-24-09098-f010:**
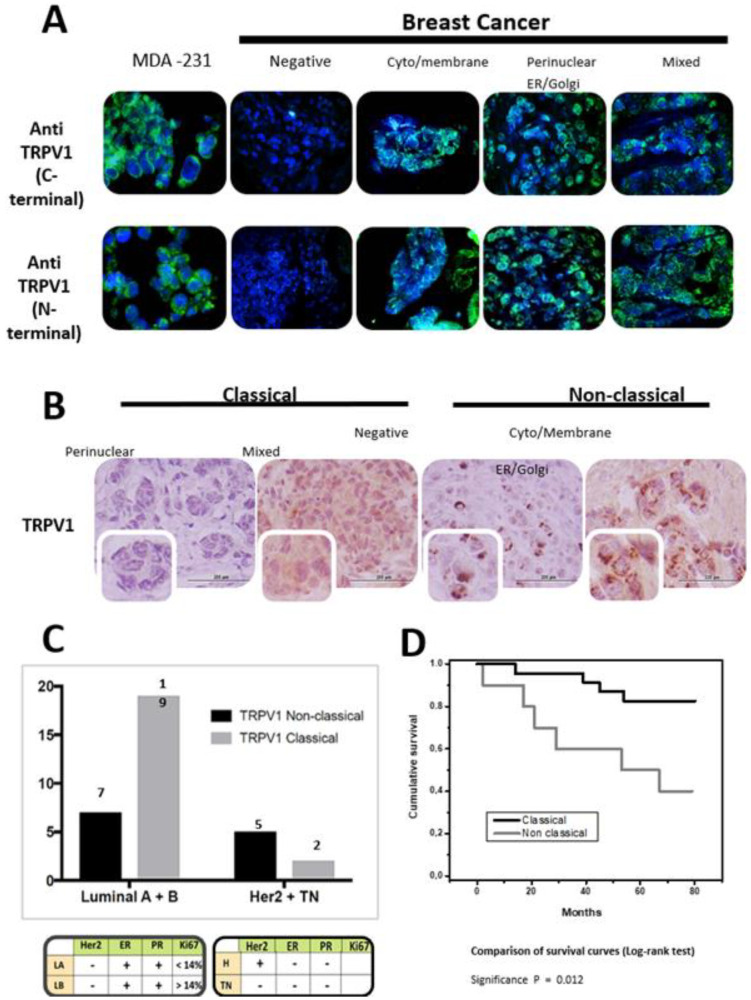
Non-classical TRPV1-like immunoreactivity patterns can identify more aggressive breast carcinomas (reproduced with permission from [196]). (**A**) The immunofluorescence of antibodies directed against the C- and N-termini of the TRPV1 protein confirmed the expression of TRPV1 in breast cancer. (**B**) The two TRPV1 paraffin immunohistochemistry staining patterns in breast cancer: “classical” in plasma membrane and cytosol, and “non-classical” with TRPV1 aggregates in endoplasmic reticulum (ER) and Golgi. (**C**) The “classical” TRPV1 staining was predominantly seen in Luminal A (LA) and Luminal B (LB) carcinoma. In triple negative (TN) cases, the “non-classical” pattern was seen more often. (**D**) Survival (Kaplan-Mayer) curves: the “non-classical” TRPV1 staining pattern was associated with worse prognosis.

**Figure 11 ijms-24-09098-f011:**
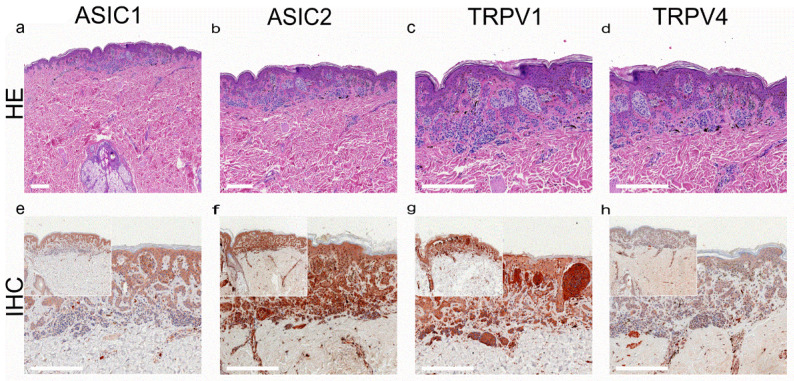
Benign melanocytic nevus of the human skin: TRPV1- and TRPV4-like immunoreactivity, along with two major acid-sensitive channels, ASIC1 and ASIC2 (reproduced from [164]). (**a**–**d**) H&E staining of tumor samples; (**e**–**h**) immunohistochemical staining (inserted smaller pictures a 2× larger perspective). Scale bars represent 200 µm.

**Figure 12 ijms-24-09098-f012:**
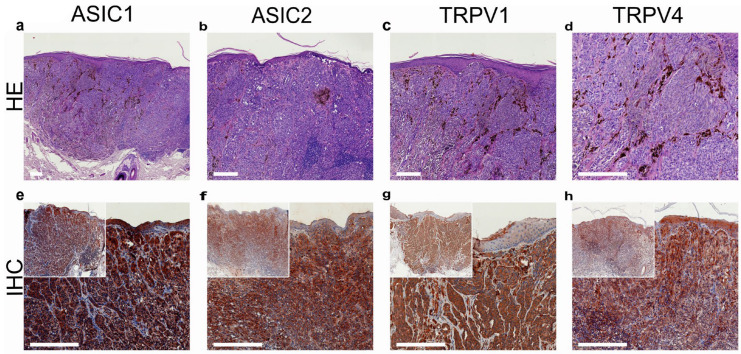
Malignant melanoma of the human skin: TRPV1- and TRPV4-like immunoreactivity, along with two major acid-sensitive channels, ASIC1 and ASIC2 (reproduced from [164]). (**a**–**d**) H&E staining of tumor samples; (**e**–**h**) immunohistochemical staining (inserted smaller pictures a 2× larger perspective). Scale bars represent 200 µm.

**Table 1 ijms-24-09098-t001:** Altered “thermoTRP” protein expression in various human cancers, and their potential use in diagnosis.

Tumor	Increased	Decreased
Oral SQCC	TRPV1, TRPA1, TRM2	
BCC	TRPV1	
Prostate cancer	TRPM8, TRPV2	
Urothelial carcinoma	TRPV2, TRPM8	TRPV1
Breast carcinoma	TRM8, TRPV6, TRPC3, TRPC6	
Gastric adenocarcinoma		TRPV1
Colorectal adenocarcinoma		TRPV1, TRPV3, TRPV4
Pancreas adenocarcinoma	TRPM7, TRPM8	
Endometrial carcinoma	TRPV2	
Ovarian carcinoma	TRPV1	
Renal cell carcinoma	TRPM2	TRPV1
Hepatocellular carcinoma	TRPV1, TRPV2	
SQCC, Squamous cell carcinoma		
BCC, Basal cell carcinoma		

**Table 2 ijms-24-09098-t002:** Altered “thermoTRP” protein expression in various human cancers, and their potential use in prognostication.

Tumor	Favorable	Adverse
Oral SQCC		TRPA1
Lung adenocarcinoma	TRPC3	TRPV1, TRPV3
Lung small cell carcinoma		TRPA1
Prostate cancer		TRPV2, TRPV6, TRPM8
Urothelial carcinoma		TRPM8
Breast cancer	TRPV2	
Gastric adenocarcinoma		TRPV2, TRPV4, TRPM8
Pancreatic adenocarcinoma		TRPM8
Endometrial adenocarcinoma		TRPV2
Ovarian carcinoma		TRPV1
Renal cell carcinoma		TRPM2

## Data Availability

Not applicable.

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
