# Peer review of "“ThermoTRP” Channel Expression in Cancers: Implications for Diagnosis and Prognosis (Practical Approach by a Pathologist)"

_ijms, 2023, doi:10.3390/ijms24109098_

Round 1

Reviewer 1 Report

The review paper by Szallasi provides an excellent overview of TPR channels with regards to cancer diagnosis and prognosis. 

I have a few suggestions for the author's consideration.

1. Abstract: suggest amending the wording 'altered expression can be used for cancer diagnosis .... and to predict outcome' to 'can potentially be used.' 

2. TRPA1, TRPM8 and TRPV4 are all expressed in the gastrointestinal tract. In addition, TRPA1 has a role in gastrointestinal mechanosensation. This would be worth mentioning.

3. I suggest adding a table to summarise the different findings with regards to TRP expression and prognosis in the different types of cancer. 

4. Conclusions: With regards to the role of the role of TRP channels in malignant transformation etc, the author refers to other reviews. It would be useful for the reader to briefly mention a few of these hypotheses in a few sentences.

Author Response

Dear Referee:

I am returning the revised version of my MS. Thank you for your suggestions for improvement. Changes are highlighted in yellow.

1) The wording of the Abstract has been changed according to your suggestion.

2) In Discussion, a paragraph has been added on TRPA1 and TRPV4 expression in the GI tract, and its therapeutic implications.

3) As per your suggestion, two Tables have been created, summarizing the diagnostic and prognostic uses of thermoTRP detection.

4) In Discussion, the postulated roles of TRP channels in carcinogenesis is now briefly summarized.

I trust that the revised MS is now acceptable for publication.

Respectfully,

Arpad Szallasi

Reviewer 2 Report

A Szallasi provides a thorough review of "ThermoTRP" channel expression across various cancers.  Szallasi gives an extensive overview of the variable protein expression in the various TRP isoforms across the spectrum of cancers.  This review could provide significant background information necessary for those who currently study or are looking to study this class of proteins in cancer.  This manuscript is suitable for publication following minor revisions, primarily focused on addressing the intersection of TRP expression and cellular metabolism. 

Comments: 

1. Because the TRP channels function as Ca2+ ion channels, adding a brief section regarding their intersection with intracellular Ca2+ regulation would be helpful to the reader. 

2. Do altered TRP channel expression affect cancer cell metabolism/mechanisms of cell death? Altered Ca2+ can impact both endoplasmic reticulum and mitochondrial homeostasis (PMID: 21798374, PMID: 33775873, PMID: 35915027, PMID: 33678551). 

3. The argument this article provides regarding TRPs as potential pathologic biomarkers in various cancers is very well explained, however, providing some additional information regarding any type of therapeutic targeting of these TRPs to enhance cancer therapy would broaden the applicability of this work.

Use of English language is acceptable.  

Author Response

Dear Referee:

I am returning the revised version of my MS. Your suggestions for improvement are much appreciated. Changes are highlighted in yellow.

1) A paragraph has been added on the role of TRPs in calcium homeostasis.

2) The postulated role of intracellular TRP channels in Ca-mediated cancer cell death is now briefly discussed.

3) The Discussion has been extended to therapeutic targeting. 

I trust that you will find the revised MS acceptable for publication.

Respectfully,

Arpad Szallasi 

Reviewer 3 Report

Dear Author

The manuscript is interesting but there are some points

1- There are 10 self-citation in the manuscript. 

2- All figures should be cited by the reference, especially pathological figures. It is not clear where are originated from who.

3- Several recent publications exist, so what is the novelty of your work? following are some of them:

-Ochoa SV, Casas Z, Albarracín SL, Sutachan JJ, Torres YP. Therapeutic potential of TRPM8 channels in cancer treatment. Frontiers in Pharmacology. 2023;14.

-Bai S, Wei Y, Liu R, Chen Y, Ma W, Wang M, Chen L, Luo Y, Du J. The role of transient receptor potential channels in metastasis. Biomedicine & Pharmacotherapy. 2023 Feb 1;158:114074.

-García-Ávila M, Islas LD. What is new about mild temperature sensing? A review of recent findings. Temperature. 2019 Apr 3;6(2):132-41.

Several English errors are in the text that should be solved

like: “ThermoTRP” expression may distinguishing distinguish between benign and malignant lesions. For example, TRPV1 is expressed in benign gastric mucosa but is absent in gastric adenocarcinoma. TRPV1 is also expressed both in normal urothelium and non-invasive papillary urothelial carcinoma, but no TRPV1 expression was seen in invasive urothelial carcinoma. “ThermoTRP” expression can also be used to predict clinical outcome outcomes.

Author Response

Dear Referee:

I am returning the revised version of my MS. Thank you for your comments, much appreciated. Changes are highlighted in yellow.

1) The number of self citations has been reduced to three (one Pharmacological Reviews and two Nature Reviews Drug Discovery papers)

2) All figures state that they have been reproduced with permission from Reference X.

3) Yes, I agree, there are a number of excellent reviews on TRP channels and cancer (many of them referenced here), but there is none written by a pathologist for pathologists. Last year, I gave a talk on thermoTRPs in cancer diagnosis and prognostication, and I came to the realization that my colleagues (academic pathologists in the leading pathology department of Hungary) have never heard of TRP channels! This prompted me to write this review. My aim was to collect all pertinent information on thermoTRP expression in cancer into one single paper. It is my hope that this paper will initiate further studies on thermoTRP expression in cancer. 

3) The English has been checked and errors corrected. 

I trust that you find the revised MS acceptabel for publication.

Respectfully,

Arpad Szallasi 

Round 2

Reviewer 3 Report

Dear Author

The manuscript is ok

there is too much self-citation

Author Response

Dear Referee:

The revised version had 4 self-citations out of 260 citations - less than 2% of the total. As per your comment, I have eliminated 2 out of the four self-citations - only two are left.

One is a "citation classic" review of mine in Pharmacological Reviews (more than 3,000 independent citations). The other is a revent, invited Nature Reviews article with three opinion leader co-authors. Hope this is acceptable. 

Arpad Szallasi